# A Qualitative Evaluation of a *Health Access Card* for Refugees and Asylum Seekers in a City in Northern England

**DOI:** 10.3390/ijerph20021429

**Published:** 2023-01-12

**Authors:** Malcolm Moffat, Suzanne Nicholson, Joanne Darke, Melissa Brown, Stephen Minto, Sarah Sowden, Judith Rankin

**Affiliations:** 1Population Health Sciences Institute, Newcastle University, Newcastle upon Tyne NE2 4AX, UK; 2Newcastle City Council, Public Health Team, Civic Centre, Newcastle upon Tyne NE1 8QH, UK; 3UK Health Security Agency North East, Health Protection Team, Civic Centre, Newcastle upon Tyne NE1 8QH, UK; 4Northumbria Healthcare NHS Foundation Trust, One-to-One Centre Shiremoor, Brenkley Avenue, Shiremoor, Newcastle upon Tyne NE27 0PR, UK

**Keywords:** refugee, asylum seeker, health access, health information, intervention

## Abstract

Refugees and asylum seekers residing in the UK face multiple barriers to accessing healthcare. A *Health Access Card* information resource was launched in Newcastle upon Tyne in 2019 by Newcastle City Council, intended to guide refugees and asylum seekers living in the city, and the professional organisations that support them, to appropriate healthcare services provided locally. The aim of this qualitative evaluation was to explore service user and professional experiences of healthcare access and utilisation in Newcastle and perspectives on the *Health Access Card*. Eleven semi-structured interviews took place between February 2020 and March 2021. Participants provided diverse and compelling accounts of healthcare experiences and described cultural, financial and institutional barriers to care. Opportunities to improve healthcare access for these population groups included offering more bespoke support, additional language support, delivering training and education to healthcare professionals and reviewing the local support landscape to maximise the impact of collaboration and cross-sector working. Opportunities to improve the *Health Access Card* were also described, and these included providing translated versions and exploring the possibility of developing an accompanying digital resource.

## 1. Introduction

In 2021, more than 89,000,000 people worldwide were forcibly dispersed from their homes as a result of persecution, conflict, violence, human rights violations or events seriously disturbing public order [1]. This included more than 27,000,000 refugees (people who have fled war, violence, conflict or persecution and have crossed an international border to find safety in another country) and more than 4,000,000 asylum seekers (a refugee whose request for sanctuary has yet to be processed) [2,3]. A total of 56,495 people applied for asylum in the UK in 2021, and of the 14,572 applications that were processed 4083 (28%) were refused [4]. A refused asylum seeker refers to a person whose asylum claim has not been granted following initial review. By the end of 2021, 100,564 asylum seekers in the UK were still awaiting an initial decision on their asylum application [4]. Mass migration on this scale has the potential to significantly impact on healthcare provision in host countries. Research suggests that displaced migrants face multiple barriers, both structural and political, to accessing healthcare in their host countries, potentially leading to unmet need and poor-quality care [5]. Although the 1951 Convention relating to the Status of Refugees and its 1967 Protocol does not explicitly define refugees’ right to healthcare, there is a more modern appreciation that access to healthcare should be regarded as a fundamental right for people seeking asylum, as is reflected in the International Organization for Migration’s 2019 Migration Governance Indicators and in the United Nation’s Sustainable Development Goals [6,7,8]. Indeed, the UN Global Compact on Refugees states that “in line with national health care laws, policies and plans, and in support of host countries, States and relevant stakeholders will contribute resources and expertise to expand and enhance the quality of national health systems to facilitate access by refugees and host communities” [9].

In the UK, the National Health Service (NHS) is a comprehensive healthcare system providing care that is free at the point of access to all UK residents. Asylum seekers awaiting review and people who have been granted refugee status in the UK are entitled to free access to all elements of NHS care. Although some restrictions are placed on the NHS care that refused asylum seekers are entitled to use, access to primary care, Accident and Emergency and 111 services (telephone triage and advice) is free and universal. People who have been refused asylum may also use services providing treatment for specific infectious diseases, sexually transmitted diseases and treatment for conditions caused by torture, female genital mutilation and/or domestic/sexual violence. Access to health visitors, school nursing and family planning services should also be freely available to this population, as should end of life care services [10].

Despite these entitlements, it is known that refugees, asylum seekers and those whose asylum application has been refused are often inappropriately denied free UK NHS care, while some individuals may not seek it due to a lack of awareness [11]. Dominant themes that emerge from qualitative studies describing barriers to care among these populations include language barriers and inadequate access to interpreter services; limited understanding of the structure and function of the NHS; difficulty meeting the costs of dental care, prescription fees, and transport to appointments; an absence of timely and culturally sensitive mental health services, properly equipped to deal with the esoteric needs of these populations; and perceived feelings of discrimination relating to ethnicity, religion and immigration status alongside concerning professional attitudes [12,13,14]. This tension, between the expectation that people seeking asylum in the UK should have access to comprehensive healthcare and the reality of the often inadequate and incomplete care that many of them actually receive, has been at the heart of efforts to improve healthcare access in these populations. However, although barriers to care are well-documented, research examining the impact of interventions intended to address and overcome them is limited: a 2021 study explored qualitative perspectives on a pre-departure medical health assessment (MHA) for refugees accepted for resettlement prior to arrival in the UK, and the Doctors of the World Safe Surgeries initiative has championed improved access to primary care services for socially excluded groups including refugees and asylum seekers [15,16]. A systematic review published in 2021 explored primary care interventions delivered primarily in North American settings and found a lack of evaluations of community-focused approaches that might be expected to be particularly beneficial in these communities [17]. Evaluations of health information resources for these populations are seldom reported. With potentially more discriminatory changes to refugee and asylum seeker policy in the post-Brexit era, and with the passing of the Nationality and Borders Bill in 2022 that effectively criminalises asylum seekers who arrive in the UK via unregistered routes, an understanding of the challenges faced by these populations in accessing healthcare and of the opportunities offered by approaches that look to address some of these barriers is more important than ever.

In response to reports that new entrants to the North East of England (including refugees and asylum seekers) were struggling to navigate the local healthcare system, Newcastle City Council’s (NCC) Public Health team, in collaboration with the Health and Race and Equality Forum (HAREF), Newcastle Council for Voluntary Services (NCVS) and the Regional Refugee Forum, developed and launched a *Health Access Card* for refugees and asylum seekers in 2019. Professional representatives from these and other organisations as well as refugees and asylum seekers living in Newcastle co-produced the resource, and 5000 copies of a folded, pocket-sized card were distributed to a number of key providers in healthcare and third sector organisations. The cards were made available in facilities commonly accessed by members of these communities (such as GP practice waiting areas and community centres), and cards were also shared with professional staff in a number of organisations to be given out in person. It was intended that the card would be used by refugees and asylum seekers themselves, and also by professionals as an adjunct to conversations concerning healthcare access. An online pdf of the card is available to view at https://cdn.cityofsanctuary.org/uploads/sites/35/2019/05/Newcastle-Health-Access-Card-2.pdf, (accessed on 4 October 2019).

The aim of this qualitative evaluation was to explore perspectives on the impact and usefulness of the *Health Access Card* through semi-structured interviews with service users and professionals based in Newcastle, and to propose improvements and changes that could be considered for future iterations of the card. A secondary objective was to use these conversations as an opportunity to explore service user and professional perspectives on refugee and asylum seeker healthcare access and on barriers to care in Newcastle more generally.

## 2. Methods

Service users with refugee or asylum/failed asylum status and professional staff who work with these groups in Newcastle upon Tyne were invited to take part in a 30–40 min semi-structured interview with a researcher from Newcastle University. There were no exclusion criteria for participation if these basic eligibility criteria were met. Previous familiarity with the *Health Access Card* was not a prerequisite to participation. Participants were recruited through partner organisations including HAREF, the Action Foundation, Refugee Voices and NCC. Researchers had intended to recruit service user participants in person, by attending community group sessions and approaching potential participants with information about the study: this approach was not feasible following the implementation of COVID-19 lockdowns in March 2020, and potential service user participants were approached instead by staff in partner organisations. Although it was originally intended that professional participants would take part in focus group discussions, semi-structured interviews were undertaken instead in view of in-person group meetings being prohibited. Participants were provided with a study information leaflet to read prior to undertaking the interview and were asked to sign a consent form to confirm their participation. The information leaflet was only available in English, but interpreters were offered in circumstances where a potential participant who did not speak English expressed an interest in taking part in the study. Ultimately, interpreting services were not used—interviews with service users were conducted in English with their agreement. Two topic guides were prepared (for interviews with service user and professional participants), exploring participant perspectives on how the *Health Access Card* had been used, on the design and content of the card, and on how the card might be improved. Participants were also asked to describe their or their clients’/patients’/friends’ experiences of healthcare in Newcastle, and to consider barriers to good care and opportunities to improve the healthcare offer for refugees and asylum seekers living in the city. Topic guides are included in the Appendix A.

The first two interviews were conducted face to face; subsequent interviews took place online on the Microsoft Teams platform or by phone in view of COVID-19 lockdown restrictions. All interviews were conducted by MM; interviews were audio-recorded with participant consent and anonymised transcripts were produced by MM, MB and SM. Interview data were coded by MM using NVivo 12 software, and a thematic analysis of emergent themes was carried out. Thematic analysis is a flexible and intuitive method of qualitative analysis that involves examining multiple data outputs for recurring patterns and motifs that can be organised into themes [18]. Codes and emerging themes and subthemes from a sample of transcripts were also discussed by the research steering group (involving all authors). Illustrative quotations are provided, and we assign a P (professional) and SU (Service User) number for each quotation.

Ethical approval for the study was granted by Newcastle University Faculty of Medical Sciences (ref: 1847/18699/2019).

## 3. Results

Eleven participants took part in this qualitative evaluation between February 2020 and March 2021. Participant characteristics are described in Table 1.

Three themes and associated subthemes emerged from the participant interview data (see Figure 1).

## 4. Barriers to Healthcare for Refugees and Asylum Seekers

### 4.1. Experiences of Healthcare

Where participants described positive experiences of using healthcare services in Newcastle, either their own experiences or those of friends, relatives or clients, these often related to interactions with individual clinical staff who had delivered tailored and compassionate care. GPs in particular were commended for some of their work with asylum seekers and refugees, and one professional participant highlighted the role that health visitors and midwives had played in arranging comprehensive care for their patients/clients. Referral to appropriate mental health support was mentioned as an example of good care:

“*We’ve had a few clients who have moved from the East end of the city over to the West end, and they won’t change their GP because the GP they’ve got is great, they love their GP and they’re wonderful and they don’t want to change, so, and that GP has been quite happy to still have that person registered with them…*”(P1)

“*Health visitors and midwives are absolutely fantastic with this client group, yeah they really go above and beyond to try to help them as much as they can, so yeah I would say that’s been really good, and I’ve done lots of joint working with midwives and health visitors…*”(P7)

Participants also described the important work done by third sector organisations in supporting refugees and asylum seekers to understand their healthcare rights, to access appropriate services, and to have recourse to non-NHS lifestyle and wellbeing support:

“*The [support organisation], that’s for LBGT support for refugees and asylum seekers, I think, yeah it is it’s on the bottom of the card, I suppose it’s not physical health but more mental health support but they’ve been really good to work with, both for the [clients] and we’ve found them really helpful as well…*”(P3)

However, largely positive experiences of healthcare for refugees and asylum seekers were not necessarily replicated in different settings or for friends/family/clients, representing inconsistent care for this population group. One service user also described frustrations when initially trying to navigate the NHS system due to unfamiliarity with UK healthcare, but these frustrations were allayed when they became more aware of NHS practices:

“*At the start I was confused, but once I got used to the system it seems alright… first time when I went to the emergency for example… I used to wait really long time [and would think] oh my goodness what’s that, but then I realised how the system works, how the patients are looked after… but yeah, it was alright…*”(SU2)

All participants described direct or indirect awareness of refugees and asylum seekers in Newcastle having poor healthcare experiences and/or struggling to access the healthcare services and support that they needed. Several participants described negative experiences of accessing or utilising healthcare services that were related to language challenges. In particular, access to interpreters and translated literature was inconsistent and frequently inadequate, and these problems were exacerbated during the COVID-19 pandemic:

“*[The client] tried to ring up to get another prescription for, I think it was sleeping pills to help her because she struggles with anxiety and they said they can’t, they can’t have a meeting with her, the GP can’t see her because they’re not doing face to face [due to COVID-19 restrictions] and they don’t have someone to be an interpreter so she’ll just have to wait until after this.*”(P3)

Negative experiences of mental health management, and of accessing appropriate mental health support, were commonly reported, and these, again, were likely exacerbated by the pandemic:

“*I’ve got this client whose daughter is having quite severe psychotic episodes, she was admitted to hospital, she was, I don’t know if it’s like a young person’s mental health team or something, she was under their care for a few weeks, and now nothing’s happened, they haven’t kind of followed up with anything, and she’s had another quite serious episode in school where she was quite dangerous to other people, and I think she’s just been, well I think they feel like she’s been left…*”(P7)

Several participants described how they or their clients/patients were treated differently by healthcare services, often in terms of presenting complaints not being appropriately investigated/explored or of being expected to wait longer for review/access to services because of their asylum seeker/refugee status:

“*They feel like that because they don’t understand, so they think because they are aliens [healthcare professionals] are reacting to them like that… “they don’t help us because we are foreigners”*”(SU1)

Experiences of problems in accessing appropriate and timely dental care were also widely reported:

“*It’s always been dental care that they’ve had most problems with, so a lot of the dentists that they go to register with tell them that they’ve got to pay, even if they’ve got an HC2, I think for some of the dental practices maybe there’s a misunderstanding with some of them…*”(P1)

Poor experiences relating to the restrictions placed on free access to healthcare services for refugees and asylum seekers were also described:

“*That lady I mentioned before who had the miscarriage, she was destitute at the time that happened and she got charged for the work they did when she had to be taken in for the miscarriage, and the letters are quite brutal in that they say that if you don’t pay this within three months we will inform the Home Office and it will affect your claim, so people really panic and she found that really upsetting, having just gone through what she went through, and I went through Doctors of the World and they said yes the charges would stand currently.*”(P1)

### 4.2. Cultural Barriers

All participants described inadequate language support as a significant barrier to care. Often, this related to difficulties in accessing services due to problems arranging appointments or understanding what services were available; in other circumstances it related to interpreters not being offered or provided, or the standard of interpreter being inconsistent or inadequate; and sometimes it related to service users being sent or provided with literature/information about their care in a language that they were unable to understand:

“*We’ve had one learner who’s sent me pictures of a letter from a consultant at a hospital to his GP, which he’s been copied into, and it’s all in English, he doesn’t speak any English, so he hasn’t received any kind of translated version of it, and it’s talking, I mean I know those letters do talk in third person, I saw so-and-so, but it’s really important the stuff that it’s talking about, and describing that there was a communication barrier and that she’s booked him in for a MRI scan anyway but she doesn’t know how much he understands…*”(P3)

Cultural variation in the way in which service users understood and interpreted health and illness was also reported as a barrier to accessing care in a UK context. This was considered particularly important in the context of service users’ often-extensive mental health needs:

“*I’m the only one who’s accessed those services [counselling], because my family doesn’t believe in mental health that much, like now they do but back then they didn’t…*”(SU3)

“*A lot of people come from cultures where mental health wouldn’t be something that was even recognised as an issue, so being able to describe that in the first place [is a barrier]…*”(P4)

For a number of participants, limited understanding of and familiarity with the UK healthcare system and its practices and procedures in terms of referrals, waiting times, etc., was seen as a barrier to refugees and asylum seekers accessing healthcare services. Where service users were perceived to access services inappropriately as a result of this, there was a feeling that this fuelled stigmatising and divisive rhetoric directed at these communities:

“*Another barrier is the assumption that people leaving their home and living here are used to the system, they assume that it’s the same everywhere but it isn’t, they’d be really frustrated if they went to my country for example and tried to access, which is not stressful for me as I understand it…*”(SU2)

### 4.3. Financial Barriers

Financial barriers to healthcare experienced by refugees and asylum seekers in Newcastle were commonly described. In some cases, this related to access to specific services for which failed asylum seekers would not be eligible for free care; in other cases, it related to the costs associated with, for example, travelling to an appointment, or, during COVID-19 lockdowns, paying for a supermarket delivery to reduce the likelihood of exposure to the virus in a public space:

“*A lot of [clients] are saying “oh well we walked for two hours to get here, if we’d paid for the bus we wouldn’t have enough food for the day” so I think in terms of weighing up often it would be prioritising is it important enough to go to the doctors, more important than buying food for the week…*”(P3)

### 4.4. Institutional Barriers

Participants described institutional barriers to care that were rooted in NHS structures and in healthcare professionals’ attitudes and behaviours. These barriers arose not only in relation to healthcare professionals’ limited awareness of the healthcare rights and entitlements of these population groups, but also in relation to the professional understanding of the particular healthcare backgrounds of refugees and asylum seekers, many of whom have considerable mental and physical health needs arising from histories of trauma and/or torture:

“*I’ve worked with lots of people over the years who’ve kind of been in and out of various medical appointments talking about physical symptoms when actually it’s turned out, you know talking about headaches talking about stomach aches talking about different things, and again some if that is their way of describing it with the language and the cultural barriers, and some of that is I think the practitioners’ awareness of the particular needs of this population and that actually, you know, lots of people have experienced trauma and persecution and different things and actually for you to be aware of that, if people are coming in with headaches or different things, just explore some of that stuff…*”(P4)

## 5. Opportunities to Improve the Healthcare Experiences of Refugees and Asylum Seekers

### 5.1. Bespoke Support

Several participants proposed offering more individualised and bespoke healthcare support to refugees and asylum seekers in Newcastle, particularly upon arrival in the city as they begin to navigate the local health and care system for the first time. Offering this population the opportunity to visit healthcare premises under the supervision of a supportive and knowledgeable guide (from the client’s accommodation provider or from a third sector/local authority organisation) was particularly well-supported:

“*Actually the most useful part was when a visitor came to my home and he actually took me and my husband to the places and he was speaking English but he tried to show how the places looked like, I didn’t understand everything he said but it gave me the idea how to get used to the system, so he actually took us to the places and, yeah, tried really hard to explain…*”(SU2)

Additional, targeted mental health support was identified as a particular priority:

“*I think like there’s more work to be done around people’s mental health, really. Like a lot more work. It’s such a huge area for our clients and for a lot of people, it’s a really difficult thing to manage and to deal with and talk about*.”(P2)

### 5.2. Language Support

Developing services that embed appropriate language support in every part of the healthcare pathway was identified as an opportunity to improve the healthcare experiences of these populations:

“*it’s [the client’s] right to be able to understand that information, so just having translated documents, and then people who understand the barriers that people might be experiencing when they’re trying to access a service, would be really useful…*”(P3)

### 5.3. Training for Healthcare Professionals

Supporting healthcare staff to be more aware of the healthcare needs and personal circumstances of refugees and asylum seekers was identified as a priority opportunity by a number of participants. Good experiences of care almost always involved compassionate and understanding professionals, recognising and addressing the sometimes-unique needs of these population groups. Facilitating a better understanding of refugee and asylum seeker healthcare entitlement, among healthcare administrators as well as clinicians, was seen as an opportunity to improve care pathways, and potentially to encourage more empathy towards service users who have arrived in the UK in challenging circumstances. There was also a feeling that unconscious biases among healthcare staff should be challenged:

“*I think it’s always a training issue. The more the surgery buys into training staff appropriately, the more awareness of someone’s situation and know what to pick up on, the work that these surgeries have been doing is great, because you know there’s a commitment there isn’t there and their staff are going to have that level of training and you know they’re kind of signed up to being welcoming to people and so, the more of that stuff that gets done the better, really.*”(P2)

### 5.4. Funding and Capacity

Several participants discussed the context in which healthcare access work with refugees and asylum seekers in Newcastle was currently funded and delivered and praised the role of the third sector in trying to support these populations’ needs in a difficult economic climate. Trying to integrate and co-ordinate some of this work across the city’s various networks was proposed as a local albeit imperfect response to ongoing challenges, in the absence of more comprehensive support from central government:

“*Because of various factors in Newcastle including years of you know austerity and funding cuts that have meant that lots of services that were previously there for asylum seekers and refugees are kind of contracted because of that, they [the voluntary sector] have been picking up asylum seekers much earlier in their journey so they’ve been picking up people who are just arriving in Newcastle, they’ve been picking up people who have been here and they haven’t had their decisions yet and actually they’ve got lots of work around integration at those earliest levels around school access and health access and different things…*”(P4)

## 6. The *Health Access Card*

### 6.1. Content and Design

Several participants commented favourably on the content of the information provided on the *Health Access Card*, mentioning in particular the useful information on the range of services available in an emergency situation and on how to access other services such as dentists and opticians:

“*Having that information on there that we can like point to, and make sure that people are informed and know how to use the kind of UK system of healthcare is really useful as well…*”(P3)

“*I think the information about the services is crucial, what service is provided at what time, and how, is important…*”(SU3)

However, one professional participant observed that the section on maternity care was potentially misleading, and should be revised and expanded:

“*I think what I’d like to see in there is something about midwives and health visitors as well, because clients don’t always realise that when you’re pregnant it’s a midwife you would see most rather than your GP, so I think that section could be improved…*”(P1)

The bright colours and effective use of images on the card was welcomed by participants:

“*It gives, like, images in its own self telling people that this is the thing you need to read if you’re looking for an optician or a dentist because there’s a picture of the glasses or a tooth, or the mother with the baby for pregnancy… so I think it’s easier in a graphical way…*”(SU3)

However, participants also felt that a lot of text on a small card might be off-putting to some people, and that the size of the card might result in it being missed or dismissed:

“*There is a lot of information on there now, obviously it’s quite dense, there is a lot of writing on there, so anybody, even if they do speak reasonably good English anybody where English isn’t a first language, it’s probably going to be quite daunting*.”(P5)

“*It doesn’t really look like something important, so… when you make something small, it doesn’t look big, the best way to say it… they don’t make [you] take it serious[ly]…*”(SU1)

### 6.2. Functionality and Distribution

Participants commented favourably on the functionality of the card, with professionals reflecting on the usefulness of the resource as a signposting tool when having discussions about healthcare with clients and service users describing the card as compact and user-friendly. For several participants, if a client or friend expressed a particular healthcare need, the *Health Access Card* was a useful physical adjunct to verbal descriptions of what was available:

“*Clients in [name of charity], yes, some of them ask us how can I access this kind of healthcare, so I just gave it to them and, you know, point them to try this…*”(SU1)

However, for some professional participants, the fact that the card was only available in English was a barrier to them and their clients/patients using it more widely, and for one service user participant, the card was a poor substitute for the more bespoke and personalised support that a support worker might give.

Professional participants also described challenges in making the resource available to its target population—if they showed a service user the card, it would often be the first time they had seen it, and there was a feeling, among professionals and service users, that the card should be made available immediately upon arrival in Newcastle to maximise its usefulness. Professional participants rarely used the card in their own practice.

“*Frankly, when I came into the country, I wanted something like this, the only thing was there was nothing at that time… now I know these places where I have to go.*”(SU3)

“*Most of the time when I’ve used it it’s been the first time that someone has seen it, I’ve never given it out and someone’s already had one or known about it.*”(P3)

### 6.3. To Improve: Language and Content

Participants recommended making the card available in other languages. Some participants also suggested that, if translated versions were not feasible, including a sentence or two in the most commonly used languages in this population that directed service users to translation and interpreting services would be beneficial:

“*Another thing is, yeah, the best option would be if it is translated into as many languages as possible…*”(SU2)

One service user also cautioned against using abbreviations and acronyms that might not be familiar to people new to the UK/NHS:

“*I don’t know what a GP is for example, I would add an explanation of this abbreviation… it’s still complicated, it’s still really hard for me, everywhere abbreviations are used and it assumes that you know about it…*”(SU2)

Participants also emphasised the importance of including information on the card that was directly relevant to the experiences of refugees and asylum seekers, in particular with regards to bespoke services (including services that promoted physical and mental wellbeing as well as healthcare services) and to accessing interpreters if and when required:

“*Maybe if there were specific projects that were aimed at refugees they might be more accessible than just the general link to the website.*”(P3)

“*Something about certain organisations that are working with people from a similar [refugee and asylum] background… so that they can tell them about the Health Access Card more properly, so that they can take them to the GP services because I know they have, like, health care champions…*”(SU3)

Participants were also keen for the card to provide some additional guidance on the nature and structure of the UK healthcare system:

“*In our countries doctors do not work like this so we just go to doctor and you can meet them right away, it’s like a drop-in, you just go there and it’s not ten minutes, so you can share whatever the problem is for as long as you want, but there are some different systems and they are expecting that to happen here but I’ve seen a lot of people, you know, complaining about how doctors are working.*”(SU1)

### 6.4. To Improve: Presentation and Format

Including more pictures on the card, to aid comprehension among service users with limited understanding of English, was suggested by several participants:

“*I think we need to make more use of images in explaining something, because images, you know, they’re international, and there’s a lot more graphic designers out there who can do, who can explain something graphically with a picture that people get the concept of…*”(P1)

“*I would like a bit more pictures, something more visual, rather than lots of writing I would add a bit more of this kind of pictures… maybe I would add a bit, for example a tiny map, for example there is not many A&E department in Newcastle…*”(SU2)

Several participants proposed making the card available online as a digital resource, which would make it easier to provide translated versions and could include expanded content and links to other online resources:

“*Maybe consider having an expanded version of the cards available online, so you could have less text on the actual card. You could say, actually you can find more information on this website…*”(P5)

However, although it was suggested that the hardware required to access online resources was often available in these communities, reliable and consistent Wi-Fi access/data provision could be challenging:

“*The majority of patients, do have [internet] access, but you, of course, there are patients that don’t either have devices or don’t have data or Wi-Fi, so whenever you are providing service online you do have make sure that is also available in other formats as well.*”(P5)

## 7. Discussion

Participants in this study described diverse accounts of the healthcare experiences of refugees and asylum seekers in Newcastle. Good experiences tended to occur on account of healthcare and other support professionals providing compassionate, personalised care, but these experiences were not consistent. Negative experiences often related to challenges in accessing care in a language that service users were able to understand, difficulties navigating an unfamiliar healthcare system, and frustrations around the availability of dental care in particular. Related to these experiences, barriers to healthcare among these populations included inadequate language support, cultural unfamiliarity with NHS structures and processes, financial barriers (including the costs of travel to healthcare premises as well as the costs of accessing health and wellbeing services themselves), and inconsistent and often-poor understanding of refugee and asylum seeker healthcare needs and entitlements among healthcare professionals. Opportunities to address some of these barriers included offering more bespoke healthcare support to refugees and asylum seekers (particularly to new arrivals to the city), embedding language support at every stage of NHS care pathways, enabling healthcare professionals to better-understand refugee and asylum seeker care needs through additional training, and working with the range of excellent third sector organisations in Newcastle to co-ordinate the important services that they provide.

The Newcastle *Health Access Card* was found to be an effective and user-friendly resource for refugees and asylum seekers in the Newcastle upon Tyne that presents helpful content in a well-designed format. The information describing emergency/urgent care services and dental provision was especially welcome, and study participants appreciated the engaging use of bright colours and graphics. Professional participants described the card as a useful *aide memoire* during conversations about healthcare with service users. However, there was a sense that the information on the card describing services that are targeted at the refugee and asylum seeker population could be expanded, and that not making the card available in non-English translations was a barrier to it being utilised more widely. It was also suggested that the density of text on the card might be off-putting to service users with limited English language skills, and that the card had not necessarily reached the service users who might benefit most from having sight of it. Recommendations for policy and practice for future versions of the *Health Access Card* include for it to be made available in other languages and to avoid abbreviations/acronyms where possible; to consider reducing the volume of text and replacing some of this with additional pictures and graphics; to expand the content describing third sector support available to these population groups; and to explore the possibility of launching a more interactive, digital resource as an alternative to the paper version.

These findings should represent a call to action for those responsible for healthcare policy and practice in the Newcastle upon Tyne to tackle the barriers to healthcare experienced by refugees and asylum seekers and to spearhead the development of inclusive, culturally sensitive and responsive services in light of the professional and service user experiences described above, but they come at a time of political turmoil and economic uncertainty. Although the war in Ukraine and the resurgence of the Taliban in Afghanistan has inspired a compassionate and generous response to refugees and asylum seekers among many parts of the UK public, media portrayals of these populations remain provocative, and it is likely that public and overt efforts to improve access to healthcare services for these groups in particular, at a time of waiting list and workforce crises in the NHS, would be highly contentious [19].

In this challenging context, a population approach cognisant of the importance of the wider determinants of health is more important than ever. A 2019 study examining refugee integration in Newcastle highlighted the social barriers to integration in a post-austerity (and pre-COVID) context, and in many instances these barriers to integration also represent barriers to effective access to healthcare resources [20]. The mental health experiences of asylum seekers in Newcastle last appeared in the research literature seventeen years ago, and many of the barriers and experiences described in 2005 are repeated in this study [21]. The author of that paper also highlights the important role that social and economic circumstances play in determining (mental) health and wellbeing. An ethnopsychiatric approach to identifying and treating mental health presentations among migrant populations, that positions mental health in its appropriate cultural context, offers a more nuanced and patient-centred response to the significant burden of mental health need described in this paper, but delivering services of this nature places demands on providers that may, in the current UK healthcare climate, be unachievable [22].

The opportunity described in this study, to map and integrate the range of services currently provided by a range of local authority, NHS and third sector organisations, so as to better understand the comprehensiveness of current support and to deliver a more joined-up and consistent offer to refugees and asylum seekers, is persuasive in this context—as in other populations, people who are economically secure and socially connected are more likely to have better health and better healthcare access. Healthcare providers should facilitate and participate in these interdisciplinary conversations and should review how the support that they offer to these populations can be improved. This may include relatively minor changes such as ensuring that clinic letters describing management plans and test results are offered in translated versions, and upskilling patient-facing and administrative staff to be more aware of these patients’ complex medical histories and healthcare entitlements. For those involved in developing health information resources for refugees and asylum seekers, this study demonstrates that there is an appetite for digital resources among these population groups, and previous research has identified a positive role for, for example, social media in supporting refugee youth to navigate and understand health systems in host countries [23]. The findings of the study also suggest that a resource that simply describes available services is potentially of less value than one that guides service users to third sector and other organisations that are able to provide more personalized and bespoke healthcare access support.

The period during which this study was undertaken, coinciding with the onset of the COVID-19 pandemic in February/March 2020 and with qualitative data collection continuing during UK lockdowns in the months that followed, served to shine a light on refugee healthcare access during crisis situations. The move away from face to face/in-person care exacerbated the barriers to healthcare already experienced by these populations, and in many instances removed meaningful access to support networks that previously would have looked to enable links to healthcare services. This had a harmful impact on service users’ physical and, in particular, mental health, and services were slow to respond to a rapidly evolving situation. These findings are in keeping with the research evidence presented elsewhere [24]. However, the shift to online provision of some services also served to act as impetus to providers and support groups to improve their clients’ digital access capabilities, and this, if sustained, offers healthcare organisations an opportunity to reconsider the means by which they engage with refugees and asylum seekers, and to look to overcome some of the barriers to care described above. The more specific experiences of refugees and asylum seekers in relation to health protection policies implemented to manage the pandemic response—including testing, quarantining and care and support for people required to isolate—were not explored in this study, but it is known that the accommodation given to vulnerable populations during the COVID-19 pandemic often fell short of providing a safe environment that was conducive to good population health management [25]. It is known anecdotally that similar challenges were faced by asylum seekers housed in temporary accommodation in Newcastle, and these experiences should be explored and documented and should inform the work of local authority and health protection practitioners in the event of future public health emergencies.

The strengths of this study include the range of professional partners involved in the research design and recruitment, the diverse sample of professional participants working in various important roles, the in-depth exploration of stories and experiences using semi-structured interviews, and the robust thematic analysis involving several members of the research team. The study has significant limitations—due to pandemic restrictions and the impact of these on recruitment, only three service participants were able to participate in semi-structured interviews, and those that did participate spoke good English and were well-established in Newcastle with settled refugee status, potentially unrepresentative of those with the most acute and urgent healthcare access needs. Several important voices, such as those of asylum seekers awaiting a Home Office decision and those of children, were not explored in this study. It is known, for example, that unaccompanied refugee minors are more likely to present with PTSD and other mental health conditions than children arriving in a host country with parents/other adult carers: the healthcare access experiences of this population are unlikely to be adequately represented in the findings of the current study [26]. Participants also had very limited experience of using the *Health Access Card* themselves or, in the case of professional participants, in their own practice, and while this is perhaps indicative of some of the challenges associated with the effective dissemination of the card, it makes any discussion of how the resource was used and the impact that it may have had impossible in the context of the current study.

## 8. Conclusions

This study sheds light on the impact of a simple but potentially wide-reaching health information resource for population groups that experience multiple complex barriers to healthcare, with important recommendations as to how the resource might be improved and expanded. It is the first study to consider the physical and mental healthcare needs of refugees and asylum seekers in Newcastle and the first to evaluate a bespoke healthcare access resource targeted at these groups, and the findings described herein are likely to be generalizable to over settings. It explores service user perspectives on barriers to healthcare alongside professional voices with extensive experience of the local and regional health and social care system, and the emerging themes complement the existing literature and offer an expanded exploration of cultural and language barriers in an urban UK context.

By virtue of the period during which interviews were conducted, the study was also able to consider refugee and asylum seeker healthcare experiences in the context of the COVID-19 pandemic. The healthcare access needs and experiences of refugees and asylum seekers newly arrived in the city who do not speak English are likely to be more extensive and complex, and future research should prioritise hearing these voices, as well as exploring the experiences of children and, in particular, unaccompanied minors. Researchers should also explore the healthcare access experiences of Ukrainian refugees, a group that was welcomed into Britain in large numbers as part of a national “Homes for Ukraine” scheme following Russia’s invasion of Ukraine in 2022, but which we anecdotally know has faced similar healthcare access challenges to those described above.

## Figures and Tables

**Figure 1 ijerph-20-01429-f001:**
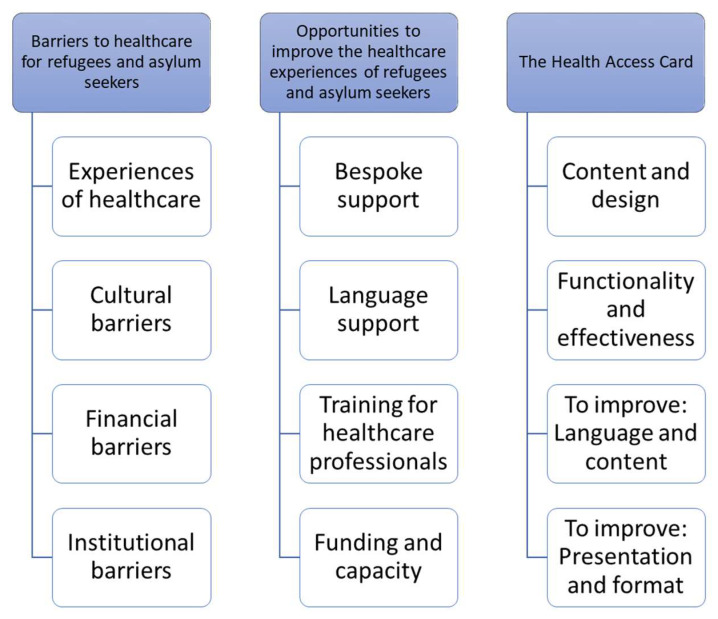
Themes and subthemes from interviews with service users and professionals around the *Health Access Card*.

**Table 1 ijerph-20-01429-t001:** Participant characteristics.

Study ID	Role	Age	Gender	Ethnicity
P (professional) 1	Third sector support worker	-	Female	White British
P2	Third sector support worker	-	Female	White British
P3	Third sector support worker	-	Female	White British
P4	Local authority support worker	-	Female	White British
P5	GP	-	Male	White British
P6	Third sector support officer	-	Male	White British
P7	Third sector organisation trustee	-	Female	Middle Eastern
P8	Local authority support worker	-	Female	White British
SU (service user) 1	Service user (refugee status)	Late 20s	Male	Pakistani
SU2	Service user (refugee status)	Mid 30s	Female	Eastern European
SU3	Service user (refugee status)	Late teens	Female	Indian

## Data Availability

The data presented in this study are available on request from the corresponding author. The data are not publicly available due to participant confidentiality.

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
