# Peer review of "A Qualitative Evaluation of a Health Access Card for Refugees and Asylum Seekers in a City in Northern England"

_ijerph, 2023, doi:10.3390/ijerph20021429_

Round 1
Reviewer 1 Report
Comments and Suggestions for Authors
Overall, this paper offers an interesting theoretical / practical reframing of “Health Access Card” for refugees and asylum seekers in a city in Northern England.
(1) I strongly suggest revising the introductory section of the paper. The authors ought to argue the ‘right and access to healthcare services’ and quote the the IOM 2019 migration governance indicators (IOM-MGIs): “(1) migrants’ rights; (2) whole-of-government approach; (3) partnerships; (4) well-being of migrants; (5) mobility dimensions of crises; and (6) safe, orderly & regular migration” (Please move statistical data to somwhere in methods section).
(2) I suggest the authors to include UN 2030 Sustainable Development Goals (SDGs) – Target 10. The IOM MGIs enhance the migration governance and sustainability nexus by clarifying the “well-governed migration” in the context of SDG Target 10.7. The research can be linked to SDG 3 Good Health and Wellbeing, SDG 8 Decent Work and Economic Growth, and so on.
(3) My suggestion would be a clear explanation of the problem as the authors see it, the tensions between policies and moral / normative imperatives, and then clarify in detail the service user and professional experiences of healthcare access and utilisation in Newcastle and perspectives on the Health Access Card.
(4) Normative aspects of health access are incorporated into EU laws and regulations. Since the post-Brexit era many legally binding regulations and directives have been amended in the UK. The amendments and new legal norms and regulations during and post-COVID era can be emphasised to make a precise comparison of pre/post-Brexit period.
(5) Despite the images of people coming on boats, most people with irregular status never did anything like that. Irregular status and irregular mobility are not the same thing. Perhaps, the authors can make a clear distinction of refugees’ statuses and classifications in their research paper. Do they use the ‘refugee’ notion according to the The 1951 Geneva/Refugee Convention (Convention Relating to the Status of Refugees) and its 1967 New York/Protocol?
(6) Finally, the authors may invoke the principle of “non-refoulement”, which is a peremptory norm in international law for asylum seekers and refugees. They may create a nexus between the right to health access and principle of “non-refoulement.” They may also support their argument by referring to the UN Global Compact for Refugees, which included binding obligations around international responsibility-sharing.
(7) I suggest looking at the conclusion’s clear argument around the converging and diverging perspectives on the Health Access Card.
Author Response
Thank you for your helpful comments. Please see our responses below:
(1) I strongly suggest revising the introductory section of the paper. The authors ought to argue the ‘right and access to healthcare services’ and quote the the IOM 2019 migration governance indicators (IOM-MGIs): “(1) migrants’ rights; (2) whole-of-government approach; (3) partnerships; (4) well-being of migrants; (5) mobility dimensions of crises; and (6) safe, orderly & regular migration” (Please move statistical data to somwhere in methods section).
Thank you. The introduction section has been updated to reference this.
(2) I suggest the authors to include UN 2030 Sustainable Development Goals (SDGs) – Target 10. The IOM MGIs enhance the migration governance and sustainability nexus by clarifying the “well-governed migration” in the context of SDG Target 10.7. The research can be linked to SDG 3 Good Health and Wellbeing, SDG 8 Decent Work and Economic Growth, and so on.
Thank you. The introduction section has been updated to reference this.
(3) My suggestion would be a clear explanation of the problem as the authors see it, the tensions between policies and moral / normative imperatives, and then clarify in detail the service user and professional experiences of healthcare access and utilisation in Newcastle and perspectives on the Health Access Card.
Thank you. The introduction section has been updated in response to this comment.
(4) Normative aspects of health access are incorporated into EU laws and regulations. Since the post-Brexit era many legally binding regulations and directives have been amended in the UK. The amendments and new legal norms and regulations during and post-COVID era can be emphasised to make a precise comparison of pre/post-Brexit period.
Thank you. The introduction section has been updated to refer to the Nationality and Borders Act and to the impact that discriminatory policies such as this are likely to have on refugee and asylum seeker health and wellbeing.
(5) Despite the images of people coming on boats, most people with irregular status never did anything like that. Irregular status and irregular mobility are not the same thing. Perhaps, the authors can make a clear distinction of refugees’ statuses and classifications in their research paper. Do they use the ‘refugee’ notion according to the The 1951 Geneva/Refugee Convention (Convention Relating to the Status of Refugees) and its 1967 New York/Protocol?
Thank you. The introduction section has been updated in response to this comment.
(6) Finally, the authors may invoke the principle of “non-refoulement”, which is a peremptory norm in international law for asylum seekers and refugees. They may create a nexus between the right to health access and principle of “non-refoulement.” They may also support their argument by referring to the UN Global Compact for Refugees, which included binding obligations around international responsibility-sharing.
Thank you. The introduction section has been updated in response to this comment.
(7) I suggest looking at the conclusion’s clear argument around the converging and diverging perspectives on the Health Access Card.
Thank you. The discussion section has been updated.
Reviewer 2 Report
I read the proposed article with great interest. It is well known that the access of refugees and asylum seekers to the health care system is an aspect that raises many critical issues that should be gradually resolved. The authors' proposal to introduce a Health Access Card is very interesting and could also serve as an example for other geographical settings
The introduction is well presented and outlines the problem well.
Unfortunately, the major limitation of this study is the procedural aspect. In order to conduct the qualitative analysis, the authors included 8 health professionals and only 3 service users in the study. It is almost superfluous to emphasize that the opinion of only three people (of different ethnicity, gender, age and probably schooling - the latter aspect is not mentioned) can by no means be considered representative. At best, it can be considered anecdotal. Moreover, the experiences they report seem to include those of other people (acquaintances or family members). Finally, with regard to health professionals, it would have been appropriate to make clear how many years they have been working in order to contextualize their statements based on their experiences. These are very important aspects that should be further highlighted in the study limitations.
However, what the authors propose highlights the refugees' perspective, and that gives this article an important meaning.
For this reason, I propose to strengthen the discussions that currently have little significance. Since the number of participants is very small, their observations can be used to broaden and generalize the discourse and address aspects in the discussions that would enrich the manuscript.
I suggest some aspects that have already been mentioned by the authors and that could be further explored, also because there are only 11 peer-reviewed articles among the references
- For example, the authors do not address refugee minors. They represent a very important aspect and are among those who need the most attention for their well-being. Why did they not consider unaccompanied foreign minors?
- refugees' mental well-being, in general, would benefit from ethnopsychiatric services, especially for victims of torture. The manuscript touches on the topic but could be better specified.
- the Covid aspect would deserve a place in the discussions. The authors mention it in the conclusion, but it deserved more space. Did refugees benefit from swabs and a health surveillance program? Because the study was conducted at the height of the pandemic, this is information from which much can be learned for the future
- What strategies could be used to make health professionals more aware of refugee health protection?
Other aspects:
- line 62: instead of "race," it would be better to use "ethnicity," as it is in Table 1
-why do the authors refer to refugees who approach physicians as "clients" rather than "patients"?
Author Response
Thank you for your helpful comments. Please see our responses below:
- Unfortunately, the major limitation of this study is the procedural aspect. In order to conduct the qualitative analysis, the authors included 8 health professionals and only 3 service users in the study. It is almost superfluous to emphasize that the opinion of only three people (of different ethnicity, gender, age and probably schooling - the latter aspect is not mentioned) can by no means be considered representative. At best, it can be considered anecdotal. Moreover, the experiences they report seem to include those of other people (acquaintances or family members). Finally, with regard to health professionals, it would have been appropriate to make clear how many years they have been working in order to contextualize their statements based on their experiences. These are very important aspects that should be further highlighted in the study limitations.
Thank you. We agree that the number of service user participants who were able to take part in this study is a significant limitation and we have reiterated this point in the discussion section. Information regarding the length of service of the professional participants was not collected at the time of interview, but this important point has been noted by the authors for future work.
- The authors do not address refugee minors. They represent a very important aspect and are among those who need the most attention for their well-being. Why did they not consider unaccompanied foreign minors?
The experiences of unaccompanied minors were not discussed or described by the participants, but it is agreed that this represents an important sub-population, and the discussion section has been expanded to reflect this.
- refugees' mental well-being, in general, would benefit from ethnopsychiatric services, especially for victims of torture. The manuscript touches on the topic but could be better specified.
Thank you for raising this point. Additional text has been added to the discussion section to expand on this point.
- the Covid aspect would deserve a place in the discussions. The authors mention it in the conclusion, but it deserved more space. Did refugees benefit from swabs and a health surveillance program? Because the study was conducted at the height of the pandemic, this is information from which much can be learned for the future.
A paragraph has been added to the discussion section to describe the pandemic experiences of professional and service user participants.
- What strategies could be used to make health professionals more aware of refugee health protection?
A response to this point has been included in the paragraph describing pandemic experiences.
Other aspects:
- 1. line 62: instead of "race," it would be better to use "ethnicity," as it is in Table 1.
- 2. why do the authors refer to refugees who approach physicians as "clients" rather than "patients"?
Thank you. The text has been amended in response to the first point. In relation to the second point, most of the professional participants referred to the people they support as clients rather than patients, but the manuscript has been updated in places to use both terms.
Reviewer 3 Report
This paper reports on a qualitative evaluation of a Health Access Card launched by Newcastle City Council in 2019. This intervention intended to provide information to refugees and asylum seekers, and professional organisations involved in supporting this population, on the healthcare services provided locally. This was in response to evidence that refuges and asylum seekers arriving in the area struggled to navigate the local healthcare system.
The authors point out the extensive literature already available reporting on migrant, refugee, and asylum seeker experiences of accessing and utilising health care services in their receiving country, including barriers to access, but equally point out the scarcity of studies reporting on the impact of interventions intended to address these inequities, particularly those targeting health literacy.
The main aim of the evaluation was to understand the impact and usefulness of the Health Access Card, and what users and professionals believed should be changed, to improve and increase the impact of the intervention. A secondary aim was to understand service users’ and professionals’ views on the experiences of refugees and asylum seekers with accessing and using local healthcare services, and barriers to access.
The process evaluation involved semi-structured interviews with three service users, and eight service providers supporting this population.
The number of participant service users is particularly small, and the authors do not address their approach to recruitment and when to stop recruitment. The findings and discussion are I think more relevant to stakeholders involved in implementing this particular intervention rather that the wider research community.
For this reason I think this paper needs major revisions to make it publication-worthy
Introduction.
The authors provide some background to the origins and development of the Health Access Card. Several questions arise: Who was involved in developing this card? Was it co-produced with refugees and asylum seekers? Several organisations are reported in the introduction such as Refugee voices but it is not clear who exactly was representing these organisations, and the process followed, including the data used to inform the development of the card.
After the cards were distributed to different organisations, how did they reach the target population? Was it left to individuals’ to pick one up or were they given it by service providers during consultations? How was the card intended to be disseminated and used?
Methods
The authors do not describe in detail their recruitment strategy.
What were the inclusion and exclusion criteria? E.g. had all those who had been approached used the card before?
Who approached potential participants? Who was in charge of giving potential participants the participant information sheet (PIS)? Who were the interpreters? E.g. were they employees of the organisations? Were they recruited specifically for this research study or were they opportunistically identified during the process of identifying potential participants? Who told service users about the study before they were given a PIS?
Why were two topic guides developed? What were the differences between the two? They should be included as supplementary material.
Which approach informed recruitment including when to stop recruitment? Were the researchers aiming e.g. for saturation, information power? The number of participant service users was small (only three participants). Were they enough to provide adequate data for analysis and drawing conclusions?
Were the interviews with service users carried out in English? What was the background of these participants? E.g. refugee, asylum seeker, in employment?
Information about participants should be under Results
Findings
The primary aim of the study was to evaluate the Health Access Card, as an intervention to improve refugee and service user experience of accessing healthcare services in Newcastle Upon Tyne. The majority of the findings however is around the barriers to access and recommendations for improving service users experience, themes that the authors have already pointed out are widely reported in the literature. Indeed, these findings reflect the findings of numerous other studies.
Findings specific to the presentation and format, and content and design of the Health Access Card would be very useful to those who have designed and piloted the card to inform future iterations, but I am not sure how much of interest they are to the wider research community. It would be more of interest to have more depth on the actual experience of using the card.
One of the excerpts from SU1 suggests this was a service provider rather than a service user – providing support to clients
Discussion
Discussion is again I think more directed to policy and decision-makers in the organisation implementing this intervention. Findings are note embedded in the literature, especially findings specific to the evaluation of the intervention. To make it relevant to the research community there should be more references to existing literature, lessons learned, and future directions, e.g. as a stakeholder interested in developing this in my own area, what should I take from this to inform my intervention?
Author Response
Thank you for your helpful comments. Please see our responses below:
- The authors provide some background to the origins and development of the Health Access Card. Several questions arise: Who was involved in developing this card? Was it co-produced with refugees and asylum seekers? Several organisations are reported in the introduction such as Refugee voices but it is not clear who exactly was representing these organisations, and the process followed, including the data used to inform the development of the card. After the cards were distributed to different organisations, how did they reach the target population? Was it left to individuals’ to pick one up or were they given it by service providers during consultations? How was the card intended to be disseminated and used?
Thank you, additional detail has been added to the text of the introduction to elaborate on the points.
- The authors do not describe in detail their recruitment strategy. What were the inclusion and exclusion criteria? E.g. had all those who had been approached used the card before? Who approached potential participants? Who was in charge of giving potential participants the participant information sheet (PIS)? Who were the interpreters? E.g. were they employees of the organisations? Were they recruited specifically for this research study or were they opportunistically identified during the process of identifying potential participants? Who told service users about the study before they were given a PIS? Why were two topic guides developed? What were the differences between the two? They should be included as supplementary material. Were the interviews with service users carried out in English? What was the background of these participants? E.g. refugee, asylum seeker, in employment?
Thank you, additional detail has been added to the text of the methods section to elaborate on these points.
- Which approach informed recruitment including when to stop recruitment? Were the researchers aiming e.g. for saturation, information power? The number of participant service users was small (only three participants). Were they enough to provide adequate data for analysis and drawing conclusions?
There was a pragmatic approach to sampling in this study, and unfortunately we were unable to recruit and interview more service user participants in view of barriers imposed by pandemic restrictions. We have added additional detail to this effect in the text. Nonetheless, the participants gave data-rich accounts and service user contributions were in keeping with professional participant accounts, from which it was possible to draw reasonable conclusions, with the caveats described in the discussion section.
- Information about participants should be under Results.
The table describing participant characteristics has been moved to the results section.
- The primary aim of the study was to evaluate the Health Access Card, as an intervention to improve refugee and service user experience of accessing healthcare services in Newcastle Upon Tyne. The majority of the findings however is around the barriers to access and recommendations for improving service users experience, themes that the authors have already pointed out are widely reported in the literature. Indeed, these findings reflect the findings of numerous other studies. Findings specific to the presentation and format, and content and design of the Health Access Card would be very useful to those who have designed and piloted the card to inform future iterations, but I am not sure how much of interest they are to the wider research community. It would be more of interest to have more depth on the actual experience of using the card.
The service user and professional participants had at-best limited experience of using the card themselves/in their own practice, and as such it was not possible to describe such experiences in the results section. This point has been added to the limitations section of the discussion.
- One of the excerpts from SU1 suggests this was a service provider rather than a service user – providing support to clients.
This particular service user did some voluntary work for the Action Foundation, and this comment is in reference to that work.
- Discussion is again I think more directed to policy and decision-makers in the organisation implementing this intervention. Findings are note embedded in the literature, especially findings specific to the evaluation of the intervention. To make it relevant to the research community there should be more references to existing literature, lessons learned, and future directions, e.g. as a stakeholder interested in developing this in my own area, what should I take from this to inform my intervention?
Thank you. The discussion section has been expanded in response to these points.
Round 2
Reviewer 3 Report
The authors have addressed comments and the manuscript has been improved considerably. My only comment is that the link in the paper to the online card is incorrect.